# Visual Function and Visual Perception among Senior Citizens with Mild Cognitive Impairment in Taiwan

**DOI:** 10.3390/healthcare10010020

**Published:** 2021-12-23

**Authors:** Chi-Wu Chang, Kuo-Chen Su, Fang-Chun Lu, Hong-Ming Cheng, Ching-Ying Cheng

**Affiliations:** 1Department of Ophthalmology, Chung Shan Medical University Hospital, Taichung 402, Taiwan; gigi78@ms16.hinet.net (C.-W.C.); jimmysu8@mercury.csmu.edu.tw (K.-C.S.); 2Department of Optometry, Chung Shan Medical University, Taichung 402, Taiwan; lu700112@gmail.com; 3Department of Optometry, Asia University, Taichung 413, Taiwan; hm_cheng@yahoo.com

**Keywords:** cognitive function, binocular vision, visual perception, developmental eye movement, peripheral awareness

## Abstract

**Purpose:** With the benefits of advanced medical technology, Taiwan has gradually changed from an aged society to a super-aged society. According to previous studies, the prevalence rate of mild cognitive impairment (MCI) over the age of 60 is 15% to 20%. Therefore, the main purpose of our study was to analyze the correlation of cognitive function with visual function (specifically, binocular vision and visual perception) in Taiwanese volunteers aged 60 years or older. **Methods:** Thirty-six healthy participants who were not taking psychiatric medications and who had not been diagnosed with any retinal or optic nerve diseases were enrolled. Addenbrooke’s cognitive examination III (ACE-III), binocular visual function, and visual perception evaluation were performed, and the data analyzed statistically by *t*-test, *χ*^2^, linear regression, and MANOVA. **Results:** Cognitive function was closely correlated with visual function and visual perception; the horizontal adjustment time of binocular eye movement, stereopsis, the motor-free visual perception test-4 (MVPT-4), and peripheral awareness actually displayed higher explanatory power in predicting cognitive function. In addition, various interactive parameters between visual function and visual perception were found to affect specific aspects of ACE-III. **Discussion:** Our study revealed that there was a close correlation of cognitive function with visual function; as such, it may be possible to predict visual function deficits in patients with mild cognitive impairment.

## 1. Introduction

Decades of medical advances in Taiwan have generally improved the health of its citizens, as evidenced by the increase in the life expectancy of the population [1]. In recent years, a special emphasis has been placed on encouraging the middle-aged and the elderly to develop lifelong learning habits. In order to strengthen their learning motivation after retirement, the Social and Family Affairs Administration of the Ministry of Health and Welfare has established senior citizen learning centers in various cities and counties. The rationale is that, aside from the basic maintenance of physical and mental wellness of the seniors, it is also important to find a way to retard the progression of dementia cases [2], so that the burden on personal and societal financial resources might be lessened.

According to previous studies, the prevalence of mild cognitive impairment (MCI) at the age of 60 or older ranges from 15% to 20% [3]. MCI progressing to dementia is at a rate of 8% to 15% per year [4]. Epidemiological studies have also shown that at least one third of MCI patients will develop dementia within 5 years [5,6]. These trends are also generally true in Taiwan [1].

Cognitive degeneration is a major sign of dementia [7]. Cognitive impairment often occurs in the elderly, and has even shown a younger trend in recent years [5]. Currently, screening for MCI in Taiwan is conducted by social workers through the use of Addenbrooke’s cognitive examination III (ACE-III, Chinese version). This test has been used for the detection of Alzheimer’s disease [8] and is proven to have a neurological basis [9]. Since mental decline appears to be part of the aging process, and is a decline in visual function, if these two operate in parallel, it may be possible to predict visual function deficits in MCI cases and implement preventive measures.

From mild cognitive impairment to dementia, patients undergoing this change may experience the simultaneous degradation of various cognitive or visual functions [10,11] or visual perception. Although these patients may not be self-aware of any difference in daily living tasks, a comprehensive assessment of impairments can often reveal memory and other visual perception deficits [12,13]. In other words, MCI may be underestimated without a comprehensive assessment. Further, although there is evidence showing that peripheral awareness has a significant contribution to maintaining postural stability [14], aging itself can decrease the binocular visual function and increase the risk of imbalance and falls [15]. A number of studies have also shown that peripheral awareness relates to the holistic visual perception [16,17].

Further, it has been found that the prevalence of binocular vision problems increases with age [18,19,20]. Understanding how visual function changes with age may predict the visual needs and functional limitations of the elderly [21,22]. In fact, the cause of oculomotor dysfunction is related to certain types of dementia [11]. In addition, it seems to manifest in saccade, esophoria, convergence insufficiency, stereopsis, peripheral awareness [16], or even in body balance. The consequences are many: it can lead to visual fatigue or reading problems [23], and all eye movement problems could increase the risk of diplopia and could result in poor performance of depth perception or fine visual motor tasks [19,24]; moreover, particularly worrisome is the increase in the risk of falls. Research has also shown that visual function provides information regarding spatial awareness, posture, and movement in response to the external environment, crucial to visuospatial perception and posture control [19]. In addition, binocular visual function might decrease because of aging and/or dementia; therefore, the degradation of cognition may have a causal relationship with the overall visual function that is yet to be explored. The main purpose of our study is therefore to investigate the decline in visual function and perception and examine the correlation with mild cognitive impairment among senior citizens in Taiwan.

## 2. Materials and Methods

A cross-sectional study was conducted during 20 November 2020 to 30 March 2021 in the CSMH affiliated Dementia Intergraded Care Center providing services and care for the elderly to prevent dementia. All the procedures were in accordance with the Declaration of Helsinki. Approval was obtained from the Institutional Review Board of the Chung Shan Medical University Hospital (Taichung, Taiwan) (Approval number: CS19110). Considering the age, physical, spirit, and compliance situation, each subject might take many different times to complete all the examination. (STROBE guideline for reporting the manuscript [25,26]).

### 2.1. Research Subjects

Healthy adults over 60 years of age who had not been identified as dementia were recruited. Exclusion criteria included stroke, brain injury, long-term use of sleeping pills or tranquilizers, and any retinopathy or optic nerve-related diseases. A total of 40 people participated initially, but four were later excluded—one with auditory nerve damage leading to poor communication, and three others had simply dropped out. The final effective number of participants was 36, with 6 males and 30 females, and their average age was 68.83 ± 9.07 years for males and 73.67 ± 9.44 years for females. According to the *t*-test analysis, there was no significant difference in age and ACE-III cognitive score in terms of gender; therefore, the results were analyzed with all 36 subjects combined, and the average age of all subjects was 72.86 ± 9.43 years.

### 2.2. Research Materials

For the cognitive status assessment, we performed Addenbrooke’s cognitive examination III (ACE-III, Chinese version), which is a detection technique that can differentiate patients with and without cognitive impairment, is sensitive to the early stages of dementia, and is available in different languages [8]. ACE-III contains 5 sub-tests, with a total of 100 questions, i.e., attention (18 questions), memory (26), language fluency (14), language (26), and visuospatial (16). The cut-off point of ACE-III is 83 [8], i.e., subjects who finish with 83 points or less are regarded as being ACE-abnormal.

Binocular visual function testing included the Howell card for distance and near phoria, Developmental Eye Movement Test (DEM), stereo acuity test (with Stereo-Fly), Ishihara pseudo-color charts for color vision, Low Contrast Flip Chart for near and distance contrast sensitivity, and Motor-Free Visual Perception Test 4th Edition (MVPT-4) and Peripheral Consciousness Card, both for visual perception. MVPT-4 had sub-tests that included visual discrimination, figure-ground, visual memory, spatial relationships, and visual closure. All tests were performed under best-corrected visual acuity. In order to avoiding any potential sources of bias, each test was performed by the same optometrist, and asked if needed rest or was tired before examination.

### 2.3. Data Analysis and Statistical Analysis

The sample size of this study was determined by using G*Power analysis, under effect size d = 0.5, α = 0.05, power (1-β) = 0.90. The calculation results of the totla sam-ple size was 36. All data were performed and analyzed by using SPSS 26.0 statistical software (IBM, Armonk, NY, USA), a value of *p* < 0.05 was considered statistically significant. (1) Independent t-test for comparing cognition, visual function and visual perception between ACE groups; (2) Pearson correlation for ACE III and MVPT-4; (3) Chi-square analysis on ACE score between four ADEM types; (4) linear regression analysis be-tween cognitive function, Visual function and visual perception; (5) Two-way ANOVA main effect comparison on Interaction between visual function, visual perception, and cognitive function.

## 3. Results

23 subjects were excluded at the beginning because of eye and mental diseases. A total of 40 people participated initially, four were later excluded, one with auditory nerve damage leading to poor communication, and three others had simply dropped out because of unable to cooperate the exam schedule. The final effective number of participants was 36 with 6 males and 30 females, and their average age was male: 68.83 ± 9.07 and female: 73.67 ± 9.44 years. According to the t-test analysis, there was no significant difference in age and ACE-III cognitive score as far as gender; therefore, the results were analyzed with all 36 subjects combined, and the average age of all subjects was 72.86 ± 9.43 years.

### 3.1. Validating ACE-III

Assessment of the validity and reliability of the ACE-III cognitive examination showed that the Cronbach’s alpha coefficient was 0.82. Based on the ACE score cut-off point of 83, a detailed analysis of the 21 ACE-abnormal and the 15 ACE-normal subjects was conducted, and *t*-test analysis further showed that the cognitive performance was significantly different between the two groups, not only in the ACE total score (*t =* −8.93, *p =* 0.000, *d =* 2.61) but also in other sub-tests, including attention (*t =* −4.54, *p =* 0.000, *d =* 1.32), memory (*t =* −9.89, *p =* 0.000, *d* = 3.34), language fluency (*t =* −5.06, *p =* 0.000, *d =* 1.71), language (t = −5.27, *p =* 0.000, *d =* 1.52), and visuospatial (*t =* −6.58, *p =* 0.000, *d =* 2.22.); these results indicated the reliability and internal consistency of the ACE-III test.

### 3.2. Correlating Cognitive and Visual Functions

Moreover, the validity of the ACE-III total score showed a high correlation (*r* = 0.741, *p* < 0.001, Figure 1) with the visual perception test (MVPT-4). It appeared that, in addition to providing effective screening for mildly cognitively impaired subjects, ACE-III could also predict the subjects’ visual perception, with good validity (*R*^2^) of 54.87%.

### 3.3. Specifics of Cognitive and Visual Correlation

#### 3.3.1. Visual Function and Cognitive Function

With the exception of contrast sensitivity and color vision, there were significant differences in eye movement (DEM) performance (including vertical time, horizontal time, and DEM ratio) and stereo vision between ACE-abnormal and ACE-normal subjects (Table 1). The ACE-abnormal group showed a significant risk of naming and combined eye movement disorder, and also displayed worse stereo vision than the ACE-normal group (Figure 2), suggesting that a decrease in cognitive abilities might be associated with the decline in eye movement and binocular fusion.

On the other hand, linear regression analysis (Table 2, Forward selection) showed that the DEM horizontal time and stereopsis had great abilities to predict cognitive performance, and the explanatory power was as high as 76.1%. The ACE total score was negatively correlated with the DEM horizontal time (*β* = −0.71, *p =* 0.000), which can be interpreted as indicating that when the horizontal time increased by 1 s, the ACE total score would decrease 0.71 points. Based on the same inference, the DEM vertical time and the DEM ratio could explain approximately 72.9% of ACE memory performance, and, in addition, the DEM horizontal time and the stereopsis could explain approximately 55.6% of ACE visuospatial perception.

In order to further assess the impact of DEM types (normal, eye movement disorder, naming disorder, combined disorder) on cognitive score, the results of the Kruskal–Wallis test showed that each ACE sub-test presented different types of DEM disorder (Table 3). Post hoc analysis showed that subjects with normal DEM abilities attained significantly higher ACE scores than the DEM combined disorder group for every cognitive dimension, while, in the ACE memory dimension, DEM normal subjects performed significantly better than those with naming difficulty and the combined disorder subjects.

Phoria tests included both distance and near examination, and the results can be classified into ortho, exo, eso, and monocular vision; the Chi square test was used to determine the correlation between cognitive function and distant and near eye positions. The results showed that there were six ortho (28.6%), five eso (23.8%), eight exo (38.1%), and two monocular eyes (9.5%) in the ACE-abnormal subjects. In the ACE-normal subjects, there were six ortho (40%), six eso (40%), three exo (20%), and no monocular vision (9.5%) (Figure 2). There was no significant correlation between cognitive function and distant phoria (χ^2^ = 3.46, *p* = 0.326). Near phoria examination found that in the ACE-abnormal subjects, there were 4 ortho (19%), 3 eso (14.3%), 12 exo (57.1%), and 2 cases of monocular vision (9.5%); and there were 3 ortho (20%), 0 eso (0%), 12 exo (80%), and 0 cases with monocular vision (0%) (Figure 3). There was no significant correlation between cognition and near phoria (χ^2^ = 4.26, *p* = 0.235), and only one subject showed a deficit in the color vision test.

#### 3.3.2. Visual Perception and Cognitive Function

The visual perception test consisted of MVPT-4 and peripheral awareness. The MVPT-4 test included visual discrimination, figure-ground, visual memory, spatial relationship, and visual closure. The results showed that ACE-normal and ACE-abnormal subjects performed significantly differently in terms of cognitive functions (Table 4).

Linear regression (Table 5, Forward selection) showed that the explanatory power of peripheral awareness and MVPT-4 visual memory for the ACE total score was as high as 71.9%; when the peripheral awareness and visual memory changed by one unit, the ACE total score would change by 0.68 and 0.38 points, respectively. Based on the same inference, MVPT-4 total score and peripheral awareness appeared to have good predictive ability (65.1%) for ACE memory.

#### 3.3.3. Interaction between Visual Function, Visual Perception, and Cognitive Function

In order to further assess the interaction between binocular vision and visual perception on cognitive function, we utilized the receiver operating characteristic (ROC) curve to group the variables; DEM, stereopsis, MVPT-4 score, and peripheral awareness were grouped and then analyzed with two-way ANOVA, which emphasized the results from the interaction first, before the main effects. For example, the ROC curve of peripheral awareness showed excellent discrimination when the cut-off point was 18.5 (AUC = 0.943, *p* = 0.000, sensitivity 86.7%, specificity 95.2%). We further analyzed the effect of interacting visual functions on various aspects of ACE, and the results showed the following.

##### 3.3.3-1. Interaction between Eye Movement (DEM) and Stereopsis on ACE Examina-tion

It appeared that DEM and stereopsis had significant interactive effects on the ACE verbal fluency (Figure 4a, *F* = 6.83, *p* = 0.014). Obviously, cognition, stereo vision, and eye movement interacted with each other and the changing visual function could even affect the fluency of speaking.

##### 3.3.3-2. Interaction between DEM and Peripheral Awareness on ACE Examination

DEM and peripheral awareness (PA) had significant interactive effects on the ACE total score (Figure 4b, *F =* 4.50, *p =* 0.042) and ACE visuospatial (Figure 4c, *F =* 4.34, *p =* 0.045), indicating that the interaction between DEM and peripheral awareness has a more significant impact on the overall cognitive and visuospatial functioning.

##### 3.3.3-3. Interaction between MVPT-4 and Peripheral Awareness on ACE Examination

MVPT-4 and peripheral awareness had a significant interaction effect on ACE visuospatial (Figure 4d, *F =* 5.00, *p =* 0.033). Main effects analysis indicated that if both visual perception and peripheral awareness presented as abnormal, the performance of cognitive ability would be worse than that with only one abnormality.

##### 3.3.3-4. Interaction between Stereopsis and MVPT-4 on ACE Examination

Stereoscopic andMVPT-4 had significant interactive effects on the language fluency (Figure 4e, *F* = 6.69, *p* = 0.014), indicating that patients with poor stereopsis and visual perception might experience an impact on language fluency.

##### 3.3.3-5. Interaction between Stereopsis and Peripheral Awareness on ACE Examination

Stereoscopic and peripheral awareness had significant interactive effects on ACE verbal fluency (Figure 4f, *F =* 4.58, *p =* 0.040) and ACE language (Figure 4g, *F =* 4.17, *p =* 0.049). On the whole, the interaction between stereopsis and peripheral awareness had a more significant impact on language and language fluency.

## 4. Discussion

The ACE-III cognitive scale was used for the screening of cognitive function in the present study. It had good reliability, with a Cronbach’s alpha value of 0.82. In addition, age and educational level also affect ACE-III cognition performance, which suggests that age and education level play an important role in cognitive function. While aging per se cannot be changed, education might have the potential to stop or reverse the course of cognitive decline.

Visual functions including eye movement (DEM), vertical time, horizontal time, DEM ratio, and stereopsis all have good explanatory power in predicting ACE cognition performance (55.5–76.1%). Previous studies had indicated that the longer the horizontal time, the higher the probability of abnormal cognitive function [21]. DEM has also been confirmed to be an important tool for evaluating cognitive function [27,28]. In addition, DEM and stereopsis have a significant interaction effect on the performance of verbal fluency. Past literature has pointed out that DEM [27] and stereopsis [29] are both closely related to the quality of language development. In the present study, most DEM-abnormal subjects exhibited both eye movement disorder and naming disorder; it can be reasonably inferred that the degradation in cognition might affect eye movement and verbal fluency at the same time. Additional literature has pointed out that there is a region in the brain connecting language and visual perception processing. This area is Brodmann area 37 (BA37) [30], which is mainly located in the fusiform gyrus and the tail of the infratemporal gyrus in the brain, and also in the medio basal and lateral sides of the tail of the temporal lobe. BA37 is also connected to the right parietal precuneus, which is Brodmann area 7 (BA7) and a part of the dorsal stream [30,31,32,33]. Therefore, the degeneration itself will affect eye movement, visuospatial, stereopsis, peripheral awareness, and even language processing [28,34]. Hence, the interruption of nerve signal connection will affect the structure of pronunciation [35].

MVPT-4 visual perception and peripheral awareness also have high explanatory power for cognition (38–71.9%). Logically, visual perception itself is a part of cognition, which organizes different information received not only by vision, but also by auditory or other sensory perception [28,30]. Our results have shown that MVPT-4 visual perception and peripheral awareness have a significant interaction with ACE visuospatial skills. Other studies have pointed out that the primary visual cortex plays a great role in the integration of visuospatial perception [27,36]. Peripheral awareness occurs when magnocellular cells signal and are stimulated by the dorsal stream, which is responsible for a part of spatial perception [28]. Clearly, early detection is an important requisite in preventing deterioration and in order for patients to return to normal life. Further, additional investigation into whether continuing education and visual training might delay the onset of cognitive impairment will be of considerable interest.

## 5. Conclusions

From mild cognitive impairment to dementia, individuals gradually feel difficulty in performing daily tasks, which can lead to a decline in their quality of life. In addition, there are also drastic changes in visual function and visual perception. Saccade, stereo vision, peripheral awareness, and visual perception are all obvious indicators in the prodromal stage of dementia. An indicator for mild cognitive impairment should place the emphasis on low-investment, more convenient, low-cost, less time-consuming, and repeatable methods. Overall, testing of visual function and perception appears suitable for the rapid screening of patients with mild cognitive impairment or pre-dementia that may lead to mobile and cognitive disabilities.

## Figures and Tables

**Figure 1 healthcare-10-00020-f001:**
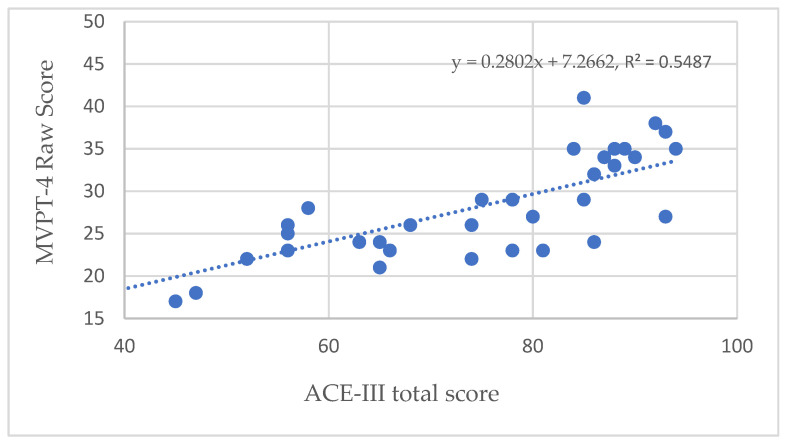
Validity of the ACE-III compared with MVPT-4.

**Figure 2 healthcare-10-00020-f002:**
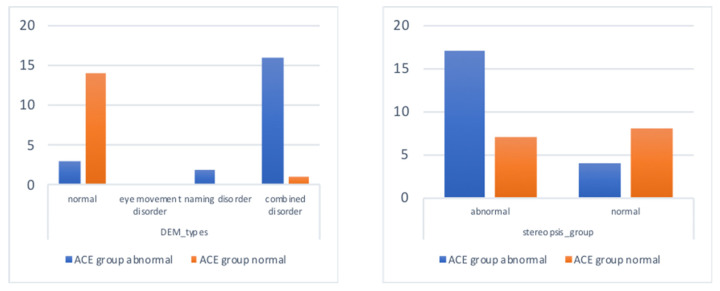
DEM and stereopsis performance between ACE-normal and -abnormal subjects.

**Figure 3 healthcare-10-00020-f003:**
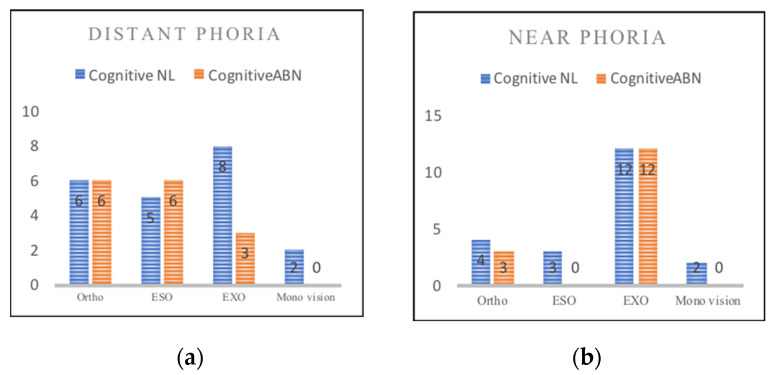
(**a**) Distant phoria in normal (NL) and abnormal (ABN) subjects. (**b**) Near phoria in normal (NL) and abnormal (ABN) subjects.

**Figure 4 healthcare-10-00020-f004:**
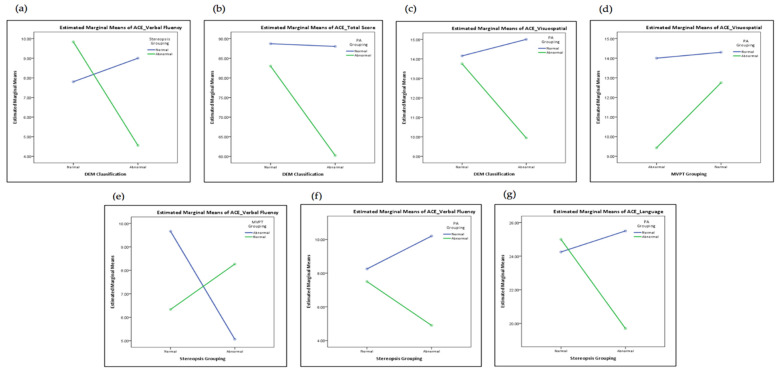
Interaction between visual function, visual perception, and cognitive function: (**a**) ACE visual fluency, DEM and stereopsis; (**b**) ACE total score, DEM, and peripheral awareness (PA); (**c**) ACE visuospatial, DEM, and peripheral awareness (PA); (**d**) ACE visuospatial, MVPT-4, and peripheral awareness (PA); (**e**) ACE visual fluency, stereopsis, and MVPT-4; (**f**) ACE visual fluency, stereopsis, and peripheral awareness; (**g**) ACE language, stereopsis, and peripheral awareness.

**Table 1 healthcare-10-00020-t001:** *t*-test analysis of visual function between ACE-normal and -abnormal subjects.

	ACE-Abnormal N = 21 M (SD)	ACE-Normal N = 15 M (SD)	Levene *F* Value	*t*	*p*	*d*
DEM VT	60.86(23.42)	30.20(6.50)	27.51	5.70	0.000	1.66
DEM HT	84.38(37.41)	31.87(8.03)	33.44	6.24	0.000	1.80
DEM Ratio	1.35(0.20)	1.06(0.12)	5.07	5.63	0.000	1.69
Stereopsis	2.40(0.55)	1.93(0.41)	4.23	2.95	0.006	0.95
Contrast Sensitivity	1.37(0.38)	1.25(0.00)	7.45	1.45	0.162	0.41

DEM: developmental eye movement; VT vertical time; HT: horizontal time.

**Table 2 healthcare-10-00020-t002:** Linear regression analysis (forward) of visual function and cognitive function.

		ACE Total Score	ACE Attention	ACE Memory	ACE Fluency	ACE Language	ACE Visuospatial
DEM VT	*β*	-	−0.78	−0.32	-	−0.66	-
	*p*	-	0.000	0.009	-	0.000	-
DEM HT	*β*	−0.71	-	-	−0.54	-	−0.65
	*p*	0.000	-	-	0.000	-	0.000
DEM Ratio	*β*	-	-	−0.62	-	-	-
	*p*	-	-	0.000	-	-	-
Stereopsis	*β*	−0.30	-	-	−0.35	−0.27	−0.44
	*p*	0.000	-	-	0.008	0.024	0.000
Adjusted *R*^2^		0.761	0.597	0.729	0.555	0.615	0.556
*F*		56.84	52.76	48.16	22.84	28.93	50.55
VIF		1.21	1	1.74	1.21	1.16	1.09

DEM: developmental eye movement; VT: vertical time; HT: horizontal time; VIF: variance inflation factor.

**Table 3 healthcare-10-00020-t003:** Chi-square analysis of ACE score between four DEM types.

	ACE Total Score	Attention	Memory	Language Fluency	Language	Visuospatial
*χ* ^2^	23.37	10.99	25.42	17.45	18.21	18.52
*p*	0.000	0.000	0.000	0.000	0.000	0.000
Post-Hoc	A > D	A > D	A > D A > C	A > D	A > D	A > D

A: normal; B: eye movement disorder; C: naming disorder; D: combined disorder.

**Table 4 healthcare-10-00020-t004:** *t*-test analysis of visual perception between ACE-normal and -abnormal groups.

	ACE-Abnormal N = 21 M (SD)	ACE-Normal N = 15 M (SD)	Levene *F* Value	*t*	*p*	*d*
MVPT	23.90(3.51)	33.60(5.57)	0.13	−7.47	0.000	2.17
Visual Discrimination	5.57(1.29)	7.67(1.18)	0.09	−4.99	0.000	1.69
Figure-Ground	4.57(1.40)	6.20(1.08)	1.33	−3.77	0.001	1.28
Visual Memory	4.95(1.36)	6.80(1.82)	2.08	−3.49	0.001	1.18
Spatial relationship	4.38(1.02)	6.07(1.67)	1.79	−3.76	0.001	1.27
Visual Closure	4.43(1.69)	6.87(1.68)	0.01	−4.27	0.000	1.45
Peripheral Awareness	13.00(4.53)	21.87(3.96)	0.15	−6.09	0.000	2.06

**Table 5 healthcare-10-00020-t005:** Linear regression analysis (Forward) of visual perception and cognitive function.

		ACE Total Score	ACE Attention	ACE Memory	ACE Fluency	ACE Language	ACE Visuospatial
MVPT-4 total score	*β*	-	0.64	0.34	-	-	0.70
	*p*	-	0.000	0.014	-	-	0.000
MVPT-4 Visual Memory	*β*	0.38	-	-	-	0.36	-
	*p*	0.000	-	-	-	0.003	-
Peripheral awareness	*β*	0.68	-	0.60	0.63	0.61	-
	*p*	0.000	-	0.000	0.000	0.000	-
Adjusted *R*^2^		0.719	0.393	0.651	0.380	0.580	0.473
*F*		45.69	23.41	33.68	22.44	25.18	32.44
VIF		1.07	1	1.71	1	1.07	1

VIF: variance inflation factor.

## Data Availability

The datasets used during the current study are available from the corresponding author.

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
