# Peer review of "Visual Function and Visual Perception among Senior Citizens with Mild Cognitive Impairment in Taiwan"

_healthcare, 2021, doi:10.3390/healthcare10010020_

Round 1

Reviewer 1 Report

This paper is well-written across the board and contributes to find the relationship between cognitive functions with visual functions. Some minor revisions are recommended for improving the quality of the paper. 

  1. In Table 3 and Table 6, linear regressions were applied on each cognitive function. How did you find optimal factors for each linear regression? (e.g., In table 3, DEM HT and Stereopsis on ACE Total score) Did you apply stepwise regression (or other variable selection methods) to find best subset on each cognitive function. Please explain detail methods you applied.
  2. In Figure 2, the Figure exceeds limit of right side of the paper. I recommend to move the Figure right into left side.
  3. Almost Tables are horizontally cut, so it is not easy for readers to understand corresponding contents clearly. I recommend to keep Tables complete by adjusting sequence of sentences for legibility.
  4. In discussion for better understanding and contribution, I recommend that you compare your results with the existing results quantitatvely in terms of comparable effects (e.g., use XX times, XX %).
  5. All statistics needs to be written italic (e.g., p → p).
  6. In conclusions, I recommend to write only your findings. Therefore, items such as importance, limitation, and/or future study (e.g., line 292 ~ 295) needs to move into the discussion.

Author Response

Response to Reviewer 1 Comments

Thank for treasure suggestions for revising,below are the new version per reviewer's question

  1. line 183,191, 235, 241 :  We applied forward selection Methods to  find best subset on each cognitive function dimension, and have revised according to reviewer's command. LINE 148, 163, 188, 194
  2. Figure 2 have been moved shift to left LINE 144
  3. ALL Tables have revised according to reviewer's command. TABLE 1-5
  4. revised, line 242 and line 259
  5. the statistics of the entire paper have been revised  
  6. Line 292 ~ 295 have moved and inserted to discussion paragraph line 266-269.

Reviewer 2 Report

Dear Author, following are my suggestions to improve your manuscript.

Abstract
Line 10: Before the aim, please include a short sentence about the research gap/relevance of your study.
Line 16 and 22-23: You can delete the information about age and education. This is generally known and therefore not an important information. Furthermore it is not a main result of your research aim.
Please include your specific research design in the methods part of the abstract.

Introduction
Line 40: The reference for MCI prevalence dates from 1999. Please delete this reference and include a recent and up-to-date reference.
Line 41: The reference about MCI progression to dementia even dates from 1984. Please delete this reference and include an up-to-date reference.
Line 42: The reference Nr. 5 is from 1988. Please delete this reference and include an up-to-date reference (if it is possible).
Line 45: Please delete this reference Nr. 5 and include an up-to-date reference.
Line 52-60: If it is possible, please delete references 10-15 and include up-to-date references. If this is a problem because of literature about visual function and visual perception, then in addition to these references you need to include current references for the other problems.
Reference 19: For risk of falls, please delete this reference and include a newer one. Maybe it is also possible to change reference 20 for a newer one?

Methods
Start first with the research design of your study and cite the reporting guideline you have used for the manuscript. (e.g. STROSA for secondary data analysis: Sunjic-Alic A, Zebenholzer K, Gall W. Reporting of Studies Conducted on Austrian Claims Data. Stud Health Technol Inform. 2021 May 7;279:62-69. doi: 10.3233/SHTI210090. PMID: 33965920: https://pubmed.ncbi.nlm.nih.gov/33965920/ or other study types e.g. STROBE, SPIRIT, CONSORT: https://www.equator-network.org/reporting-guidelines/spirit-2013-statement-defining-standard-protocol-items-for-clinical-trials/ ) Check your content with the appropriate  guideline to include further missing information’for the entire manuscript.

Results
Start first with the characteristics of the sample – see reporting guideline.

Discussion:
Where possible, please replace your older literature by up-to-date literature.
Include the limitations of your study.

Author Response

Response to Reviewer 2 Comments

Thanks for reminding the problems of the manuscript

Abstract
1. Line 10: Before the aim, please include a short sentence about the research gap/relevance of your study. 

responds: revised, line 10-12

2. Line 16 and 22-23: You can delete the information about age and education. This is generally known and therefore not an important information. Furthermore it is not a main result of your research aim.

responds: revised, line 15-18

3. Please include your specific research design in the methods part of the abstract. 

responds: revised, line 18

Introduction to discussion

responds: all the reference have been revised and update to recent reference, see the red number in the paragraph and literature references.

Round 2

Reviewer 2 Report

Dear author, please revise your manuscript as suggested in the last review. Thank you!

Methods
Start first with the research design of your study and cite the reporting guideline you have used for the manuscript. (e.g. STROSA for secondary data analysis: Sunjic-Alic A, Zebenholzer K, Gall W. Reporting of Studies Conducted on Austrian Claims Data. Stud Health Technol Inform. 2021 May 7;279:62-69. doi: 10.3233/SHTI210090. PMID: 33965920: https://pubmed.ncbi.nlm.nih.gov/33965920/ or other study types e.g. STROBE, SPIRIT, CONSORT: https://www.equator-network.org/reporting-guidelines/spirit-2013-statement-defining-standard-protocol-items-for-clinical-trials/ ) Check your content with the appropriate  guideline to include further missing information’for the entire manuscript.

Results
Start first with the characteristics of the sample – see reporting guidelines.

Author Response

Dear reviewer, thanks for your suggestion, and have revised according to the guideline. 

Round 3

Reviewer 2 Report

the authors should write in Materials and Methods that they used the STROBE guideline for reporting and they should cite the guideline as suggested in my last reviews

This manuscript is a resubmission of an earlier submission. The following is a list of the peer review reports and author responses from that submission.